# Ontogeny of the Cytochrome P450 Superfamily in the Ornate Spiny Lobster (*Panulirus ornatus*)

**DOI:** 10.3390/ijms25021070

**Published:** 2024-01-15

**Authors:** Courtney L. Lewis, Quinn P. Fitzgibbon, Gregory G. Smith, Abigail Elizur, Tomer Ventura

**Affiliations:** 1Centre for Bioinnovation, University of the Sunshine Coast, 4 Locked Bag, Maroochydore, QLD 4558, Australia; cll014@student.usc.edu.au (C.L.L.); aelizur@usc.edu.au (A.E.); 2School of Science and Engineering, University of the Sunshine Coast, 4 Locked Bag, Maroochydore, QLD 4558, Australia; 3Institute for Marine and Antarctic Studies, University of Tasmania, Private Bag 49, Hobart, TAS 7001, Australia; quinn.fitzgibbon@utas.edu.au (Q.P.F.); gregory.smith@utas.edu.au (G.G.S.)

**Keywords:** Cytochrome P450, *Panulirus ornatus*, transcriptomics, ecdysteroidogenesis

## Abstract

Cytochrome P450s (CYP450s) are a versatile superfamily of enzymes known to undergo rapid evolution. They have important roles across growth and development pathways in crustaceans, although it is difficult to characterise orthologs between species due to their sequence diversity. Conserved CYP450s enzymes in crustaceans are those associated with ecdysteroidogenesis: synthesising and breaking down the active moult hormone, 20-hydroxyecdysone. The complex life cycle of the ornate spiny lobster, *Panulirus ornatus*, relies on moulting in order to grow and develop. Many of these diverse life stages have been analysed to establish a comprehensive transcriptomic database for this species. The transcripts putatively encoding for CYP450s were mapped using transcriptomic analysis and identified across growth and development stages. With the aid of phylogeny, 28 transcripts of 42 putative *P. ornatus* CYP450s were annotated, including the well conserved Halloween genes, which are involved in ecdysteroidogenesis. Expression patterns across the life stages determined that only a subset of the CYP450s can be detected in each life stage or tissue. Four *Shed* transcripts show overlapping expression between metamorphosis and adult tissues, suggesting pleotropic functions of the multiple *Shed* orthologs within *P. ornatus*.

## 1. Introduction

Cytochrome P450s (CYP450s) are an ancient, rapidly evolving and versatile superfamily of monooxygenase enzymes [1]. The genes encoding for these enzymes are one of the largest gene superfamilies in plants, animals and fungi and are essential for a variety of reactions within enzymatic pathways of key molecules required for development, growth and reproduction [2,3]. CYP450s get their name from the absorption peak at 450 nm, which is caused by their heme group (a porphyrin ring which stabilizes an iron atom) that is bound to carbon monoxide and is characteristic of all enzymes in this superfamily [4]. The genes in this group are known to cluster within the genome, which may be the cause of such elevated evolutionary change and duplication rates [2]. Their rapid evolution means that there is a continual change in the number of identified CYP450s [5], and even between closely related species, numbers of CYP450s differ, due to deletions or expansion in the number of genes [1,2]. The significant CYP450 sequence diversity, with minimal conservation, leads to the challenge in characterising this enzyme superfamily. Within crustacean CYP450s, there are significant knowledge gaps concerning their structure, function, and regulation [6]. A significant portion of initial hypotheses for their roles in crustaceans were taken from research of insect CYP450s, as the closest evolutionary relatives with characteristic similarity; however, there are also significant differences between crustaceans and insects [7]. This variation is exemplified in the functionally conserved CYP450s named *Shade* in insects and *Shed* in crustaceans; the different naming is owing to their conserved function albeit distinct phylogeny [2].

Conserved CYP450s in crustaceans are involved in ecdysteroidogenesis, the enzymatic pathway which generates the moult hormones [8]. Moulting is used by arthropods, including insects and crustaceans, to allow for growth and morphological changes [9]. This developmental process occurs when the animal sheds their hardened exoskeleton and forms a new and larger exoskeleton [10], as regulated by fluctuations in the concentration of ecdysteroids [11] across four stages of the moult cycle: premoult, ecdysis, postmoult and intermoult [12]. This is a well-characterised pathway as it is conserved across arthropod species and was identified in crustaceans through phylogenetic studies with other arthropod species including insects, where moulting is well-researched. The biosynthesis of the active moult hormone 20 hydroxyecdysone (20HE) relies on the sequential activity of a group of Cytochrome P450s (CYP450s), together termed the Halloween genes (Figure 1). The majority of the ecdysone synthesis pathway occurs in the Y-organ [13], as the site responsible for the coordination of the moult pathway [14]. The initial steps of this conversion process are regulated through *Spook* (CYP307A1) and the paralogs *Spookier* (CYP307A2) and *Spookiest* (CYP307B1) [15]. The enzyme *Phantom* (CYP306A1) is responsible for the next conversion step in the pathway, followed by a series of enzymatic reactions catalysed by *Disembodied* (CYP302A1), *Shadow* (CYP315A1), and *Shade* (CYP314A1) [16]. The final conversion from the precursor ecdysone to the active 20HE is catalysed by *Shade* and occurs in the target tissues [17]. Arthropod CYP clan diversity is lower than that of plants, fungi and vertebrates, with six well supported clans: CYP2, CYP3, CYP4, CYP16, CYP20 and mitochondrial [3]. CYP2, CYP3 and CYP4 are the major clans represented in arthropods. The enzyme CYP18A1, a member of clan 2, assists with the degradation of 20HE [2]. Whilst the Halloween genes and their roles in synthesising ecdysone have been well established, there is a gap in our understanding of the pathway preceding the Halloween genes, with the initial reactions involved characterised as a “Black box” due to their elusiveness [18]. This process is critical for the growth and development of all crustaceans, including spiny lobsters, which have a complex life cycle comprising multiple morphologically distinct life stages. By examining expression patterns across the varied life stages, tissues and moult stages, it is possible to infer putative functions, highlighting candidate CYP450s that may have a role in the “Black box”.

Spiny lobsters begin their life as a tiny egg that hatches into a transparent, leaf-like organism called phyllosoma [19]. This is an oceanic phase, which can last up to two years in some species [20], during which the animal may be entrapped in water currents known as gyres, where they actively predate on zooplankton to supply their nutrition for growth [21]. Following multiple moults (up to 24 moults in *Panulirus ornatus*) and accompanying growth, the phyllosoma transforms into the non-feeding puerulus, a transparent and miniature version of the adult lobster [22]. The puerulus actively swims from off the continental shelf to reach the coastal benthic habitat of adult spiny lobsters in a journey that can be hundreds of kilometres. As the puerulus does not have digestive organs, it must make this journey without eating [23]. This journey is fuelled by energy stores in the form of phospholipids, accumulated during the previous larval phase [21]. Once the animal reaches the benthic habitat, the puerulus moults into the benthic juvenile, which is spiny and pigmented, resembling a miniature version of the adult lobster. It must resume feeding to rebuild its energy stores, which were depleted during the puerulus journey and metamorphosis, to sustain its vital functions and enable further growth through continual moulting [24].

Spiny lobsters are important globally, forming the basis for commercial fisheries in multiple countries in both tropical and temperate zones. Commercial fisheries rely on natural puerulus settlement and the subsequent harvest of wild populations, and without the conservative fisheries management, the industry faces the risk of decline [25]. Establishing sustainable aquaculture techniques is a desirable method to reduce the reliance on wild populations and to meet the increasing commercial demand [26]. Whilst some spiny lobster species have characteristics ideal for culturing, their multiple morphologically distinct stages with life-stage-specific nutrient requirements [22] and extended larval duration makes rearing them from egg to product in captivity a unique challenge [27]. The genus *Panulirus* is the largest group within Palinuridae, composed of forty-nine species that are highly sought after in multiple countries [28,29]. In particular, the tropical, ornate spiny lobster (*P. ornatus*) is considered a prime candidate for aquaculture due to its shorter larval phase, rapid growth and high commercial demand and value in the marketplace [30].

In insects, the final conversion step of the moult hormone 20HE occurs within the target cells in the peripheral tissues and is catalysed by an enzyme named shade [31], which was only recently identified in crustaceans. A study on the sub-temperate eastern spiny lobster, *Sagmariasus verreauxi,* identified 43 putative CYP450s, with phylogenetic annotation available for 8 of them [2]. Of the 43, 11 displayed variations in expression across metamorphosis, with four dominant expression patterns [2]. The same study evaluated a particular group of CYP450s thought to be functioning as shade, through an enzyme assay carried out in transfected COS-7 cells. This study determined that these CYP450-encoding genes share a function with shade, however they are phylogenetically differentiated and therefore, the enzymes were named *Shed* [2]. Apart from the Halloween genes and those associated with ecdysteroidogenesis, the CYP450s in crustaceans remain largely unknown.

A comprehensive transcriptome is available for *P. ornatus*, detailing samples obtained from eleven embryo stages [32], twelve distinct metamorphic stages ranging across the lobster life cycle from actively predating phyllosoma, to the non-feeding puerulus, to the predating juvenile [33], and 52 additional tissue samples obtained for 17 tissues from adult and juvenile *P. ornatus* [34]. This is a valuable resource that can be used for the identification of cytochrome P450s within the species across the life stages and particularly how they change in expression across metamorphosis and the tissues. Previous transcriptomic studies focusing on the Y-organ of decapod crustaceans have uncovered key genes, including CYP450s, associated with moult pathway regulation [35]. The expression of ecdysteroidogenesis related genes can help to predict other CYP450s that may be part of ecdysteroidogenesis. It is likely that CYP450s without any current characterisation play a role in early ecdysteroidogenesis, potentially as part of the elusive “Black box”.

This research aims to characterise the CYP450s identified within the *P. ornatus* transcriptome, providing valuable insights into mechanisms and pathways within the species, and how it has diverged from related arthropods. Phylogeny with closely related decapod species including *S. verreauxi* will assist in providing annotation to known CYP450s, highlighting those involved with ecdysteroidogenesis and juvenile hormone biosynthesis. Genes will be mapped across the life stages and tissues to identify genes with differential expression to identify CYP450s with key roles across the *P. ornatus* life cycle.

## 2. Results

### 2.1. Phylogenetic Analysis

A PFAM domain search for a CYP450 domain across the predicted proteins derived from the *P. ornatus* transcriptome [33] retrieved the putative CYP450s (n = 116). Following the selection of transcripts encoding predicted proteins with a minimum of 300 amino acids and removal of redundancies (identical open reading frames), 42 putative CYP450s were identified, ranging in size from 306 to 622 amino acids. Multiple sequence alignments followed by phylogenetic analysis with CYP450s from arthropod species, including the closely related *S. verreauxi*, annotated 28 CYP450s (Figure 2). Three paralogs were identified for CYP302A1 (*Disembodied*), and six paralogs were identified as *Shed*. Other annotated CYP450s include a CYP18A1 ortholog and two CYP15A1 orthologs (Table 1). From phylogenetic analysis with a collection of related arthropods, supported by protein sequence analysis, a total of 28 *P. ornatus* transcripts were assigned putative annotations. The neighbour-joining tree generated nine distinct clusters with orthologues of CYP4c15, CYP307 (*Spook*), CYP18A1, CYP306 (*Phantom*), *Shed* (found in decapods), CYP15A1, CYP315A1 (*Shadow*), CYP302A1 (*Disembodied*) and CYP314 (*Shade*, found in insects and branchiopods).

### 2.2. Differential Gene Expression across Panulirus ornatus Life Stages

Gene expression of the 42 putative CYP450s was analysed across 11 stages of embryogenesis, 12 stages across metamorphosis, the hepatopancreas across 6 stages of the moult cycle in juveniles (n = 3 RNA-Seq libraries per stage, a total of 87 RNA-Seq libraries), and tissues from male and female juvenile and adult *P. ornatus* (a total of 52 RNA-Seq libraries representing 17 tissues). Twenty-five CYP450s were identified in the embryos and mapped across 11 stages of a 30-day period of embryo development (Figure 3). Ten CYP450s were upregulated in day 0 embryos, including CYP3A2-like, CYP307A1 (*Spook*), and CYP302A1d (*Disembodied*). There were four upregulated CYP450s in day 3 embryos, including two *Shed* orthologs. Of the annotated CYP450s, only CYP18A1 showed differential expression across the mid-stages of embryo development, with elevated expression from day 15 to day 21. There is limited CYP450 expression in the later embryo stages prior to hatching on day 30.

Digital expression patterns highlighted CYP450s that varied in expression across the 12 stages across metamorphosis. Of the 42 transcripts encoding putative CYP450s, 27 displayed distinct differential expression (Figure 4). The expression levels of each transcript were compared to identify similar patterns between transcripts and trends in expression across the distinct life stages. Following analysis, the transcripts were categorized into four expression profiles: upregulation in phyllosoma, upregulation in the puerulus, upregulation at the first instar juvenile, and upregulation at four days following the juvenile transition. *Shed B*, *Shed C*, and *Shed E* are upregulated in the phyllosoma. CYP4G1 is one of three transcripts with elevated expression in the puerulus. CYP15A1b and CYP4c15 are upregulated following the juvenile transition.

Of the 42 putative CYP450s identified, 28 were identified in the hepatopancreas across a juvenile moult cycle, 10 of which showed specificity to the hepatopancreas including *Shed F*, CYP15A1, CYP4, CYP11A1 and CYP2L1. Four transcripts demonstrated higher expression from J7-6 to J7-12, including *Shed D* and CYP15A1 (Figure 5A). Expression of CYP450s is highest in the hepatopancreas of postmoult animals, with lower expression throughout the intermoult period. This expression profile was evident for nine transcripts, of which seven were hepatopancreas-specific, including CYP2L1 and *Shed F* (Figure 5B). The remaining 14 CYP450s identified in the hepatopancreas had very minimal expression across the moult cycle. Annotated CYP450s that were not identified in the hepatopancreas include *Shed C* and CYP306A1 (*phantom*).

In the adult *P. ornatus* tissues, almost all the putative CYP450s demonstrate differential expression (see Appendix A), with upregulated expression in the hepatopancreas for 28 of the putative CYP450s, including several *Shed* orthologs, CYP15A1b, CYP4b and CYP4c, and CYP3A2-like. The annotated CYP450s involved in the biosynthesis and degradation of 20HE were targeted to observe their expression across the larval stages and the adult tissues of *P. ornatus*. Across the metamorphic life stages, four putative *Shed* transcripts demonstrate elevated expression in two expression profiles (Figure 6A). *Shed B* and *Shed E* show increased expression throughout the phyllosoma stages, whilst *Shed C* and *Shed F* have increased expression in the juvenile stage. The same four shed transcripts also have the highest expression in the adult tissues, although their expression profile grouping is different. *Shed B* and *Shed C* share an expression pattern that is quite broad across the tissues with expression evident in most tissues, with elevated expression in the neural tissues and hepatopancreas. In contrast, *Shed E* and *Shed F* expression is almost exclusive to the hepatopancreas (Figure 6B).

### 2.3. Uncharacterised CYP450s across Life Stages and Tissues of P. ornatus

Across all life stages and tissues, there were 14 unannotated CYP450s with differential expression across the life stages (Table 2). These CYP450s were mapped across embryo development, metamorphosis, hepatopancreas across a juvenile moult cycle, and 17 tissues from male and female adults (Figure 7). Of those differentially expressed genes, several were differentially expressed across different life stages; Cluster-1222.197145 was differentially expressed in both embryo development and larval development, and Cluster-1222.108378 and Cluster-1222.115281 were both differentially expressed in larval development, hepatopancreas across moult stages, and adult tissues. The 12 differentially expressed genes were given putative annotation based on amino acid sequence similarity with known CYP450s from arthropod species.

## 3. Discussion

Cytochrome P450s are a versatile superfamily of enzymes with diverse roles that are known for their rapid evolution and expansion [1]. This creates a challenge when identifying orthologues between species. A CYP450 PFAM domain search retrieved 116 potential CYP450 encoding transcripts, and followed by selection for a minimum of 300 amino acids, a total of 42 CYP450s were identified across the life stages of *P. ornatus*, ranging in size from 306 to 622 amino acids. Phylogenetic analysis with a collection of arthropods, including the closely related *S. verreauxi*, inferred annotation to twenty-eight CYP450 encoding transcripts, including the conserved Halloween genes, involved in ecdysteroidogenesis. The number of Halloween genes identified in *P. ornatus* is in keeping with results obtained in other decapods, with three CYP302A1 paralogs and six *Shed* paralogs; however, it differs from insects [2,16]. The *P. ornatus* Halloween genes cluster tightly within their respective enzyme subclass, apart from the six *Shed* transcripts, which are separated from shade by several clusters of CYP450 enzyme families. This validates the considerable difference between *Shade* in insects and *Shed* in crustaceans. Even though they are equivalent genes performing the same function, catalysing the final conversion step to 20HE in the target tissues of the animal [2], they are very different in their protein sequences. Other annotated CYP450s include CYP18A1, a key enzyme in invertebrates responsible for the degradation (and thereby inactivation) of the moult hormone [41], and two transcripts identified as CYP15A1, the enzyme that catalyses the epoxidation of methyl farnesoate to juvenile hormone [42].

The *P. ornatus* CYPome comprises seven genes from the CYP2 clan, two genes from CYP3, five genes from CYP4, and fourteen genes belonging to the mitochondrial clan, with no genes from the CYP16 and CYP20 clans. Enzymes belonging to CYP2 are diverse and have roles in a range of physiological processes [43]. In *P. ornatus*, the CYP2 clan includes CYP307A1, CYP306A1, CYP18A1, CYP15A1, which are all associated with ecdysteroid metabolism, as well as CYP1A1-like, which is linked to the metabolism of steroid hormones, including estrogen in vertebrates [44]. The CYP3 clan in *P. ornatus* contains CYP3A2 and CYP6a14, and this clan is associated with detoxifying xenobiotics and endobiotics, which are responsible for the metabolism of steroid hormones in vertebrates [45]. CYP306A1 demonstrates high expression in the reproductive tissues of other decapods [46], and is expressed highly in some of the adult testis and ovaries of *P. ornatus*. CYP4 genes are highly abundant in arthropod genomes and are implicated in fatty acid and xenobiotic metabolism [45], with CYP4c15, CYP4G1, CYP4 all part of the CYP4 clan in *P. ornatus*, and are implicated in the pathways of ecdysteroidogenesis, lipid metabolism and omega hydroxylation, respectively. The mitochondrial clan is typically small and includes genes involved with steroid metabolism [47]. Unexpectedly, this clan was the most abundant in *P. ornatus*, including six *Shed* transcripts as well as CYP315A1, CYP302A1, CYP44 and CYP11A1, the majority of which are involved in steroid metabolism.

The CYP450 transcripts’ gene expression was mapped across the extensive transcriptomic dataset established for *P. ornatus* at the University of the Sunshine Coast, and available on CrustyBase [48], encompassing 11 stages of embryogenesis [32], 12 stages of larval development [33], the hepatopancreas across 6 moult stages in juveniles (generated in Chapter 4), and tissues from adult *P. ornatus* [34]. All 25 CYP450s detected in embryo development show distinct differential expression, with minimal overlap in the genes expressed on each day of embryogenesis. Ten of the CYP450s were expressed only in day 0 embryos, including CYP302A1d (a *Disembodied* paralog), *Spook*, and CYP3A2-like. It is possible that transcripts with expression specific to day 0 of embryogenesis are linked with initialising the embryonic development and cell division as the day 0 embryos were collected at the 4–8 cell stages [32]. Then there is a group of eight different CYP450s with expression from day 3 to day 9, including *Shed B* and CYP302A1a. The genes with high expression on these days are likely to be implicated in processes such as gastrulation, where the germ layers appear, which occurs on day three [32], followed by segmentation and formation of the nervous system and organs [49]. This is followed by a set of eight different CYP450s expressed from day 12 to day 24, which include CYP2L1-like, CYP18A1, and *Shed C*. Both CYP18A1 and *Shed C* are essential for the conversion of 20HE [41], and their expression across the mid-stages of embryogenesis suggests 20HE may be implicated in the significant structural changes taking place, following the segmentation and establishing the body structures. Finally, CYP4c15 is specific to day 27, where it is significantly upregulated. This CYP450 is implicated in ecdysteroid biosynthesis [50], and its expression may suggest the embryo is getting ready for hatch, which occurs on day 30, and the subsequent transition into the brief nauplius stage following hatch [51]. CYP450s are known to be essential for arthropod development; mutations in the important Halloween genes result in significant embryo deformities and mortality [2]. Across the 30-day embryogenesis period, there is a sequence to the expression of CYP450s with one cluster of 10 CYP450s expressed in day 0, followed by the expression of 6 different CYP450s from day 3 to day 9, followed by 8 CYP450s with expression across day 12 to day 24. This demonstrates important developmental functions of CYP450s, and it is likely that putative CYP450s without annotation also have essential roles, yet to be uncovered.

Across metamorphosis, 27 of the 42 CYP450s demonstrated distinct differential expression. Gene expression fluctuations across the spiny lobster life stages are expected due to the significant genetic and physiological changes that occur between the disparate life stages, with up to 25% of the transcriptome changing across metamorphosis [22]. The transcripts were placed into four expression patterns, following similar results in *S. verreauxi*, where four dominant expression patterns were identified in 11 differentially expressed CYP450s across five metamorphic life stages [2]. Phyllosoma grow through a series of moulting events, eventually undergoing metamorphosis to a puerulus [52]. This is an energetically demanding process that requires the phyllosoma to amass sufficient energy stores [53]. The upregulation of CYP450 gene expression during this stage indicates these genes may be important for the processes of digesting, transporting and storing nutrients for use in the later life stages. *Shed C*, *Shed E* and CYP18A1 fall into this category, although they are known to also be involved in 20HE biosynthesis. This expression is likely due to the several moulting and metamorphic events that occur throughout larval development. The second expression profile included a CYP4G1 orthologue, an enzyme putatively associated with lipid metabolism [31]. The CYP450s with upregulated expression in the first instar juvenile may be important for recovery post-juvenile transition, such as the mobilisation of lipids to restore the depleted metabolic reserves. Thirteen CYP450s are upregulated in the juvenile four days following the juvenile transition, including CYP15A1 and CYP4c15. CYP15A1, the enzyme that catalyses the conversion of methyl farnesoate to juvenile hormone [42], is upregulated at this stage. Previous studies have suggested CYP4c15 is implicated in ecdysteroidogenesis [54].

CYP2L1 was the first marine invertebrate CYP450 to be identified, isolated from the hepatopancreas of *Panulirus argus* [6]. Since then, there have been several studies investigating CYP450s in the hepatopancreas across crustaceans [55,56] and CYP2 and CYP3 clans have been confirmed as abundant in crustacean hepatopancreas [6]. This suggests that CYP450s have fluctuating expression in the hepatopancreas in accordance with other changes relating to development stage, diet, environment, and moulting. A total of 10 of the 28 CYP450s identified in the juvenile *P. ornatus* hepatopancreas showed specificity to the hepatopancreas in adult *P. ornatus*, including *Shed F*, CYP15A1, CYP4, CYP11A1 and CYP2L1. Across the J7 to J8 moult period, CYP450 expression was generally higher in the postmoult hepatopancreas, with only a few CYP450s, including *Shed D* and CYP15A1, upregulated in the intermoult hepatopancreas. The majority showed very limited expression in the hepatopancreas across the moult cycle. In the blackback land crab, *Gecarcinus lateralis*, six *Shed* transcripts were detected in the hepatopancreas, with one paralog, *Gl-Shed2*, having significantly higher expression in the hepatopancreas [57]. In the *P. ornatus* hepatopancreas, four *Shed* paralogs were expressed. *Shed C* and *Shed F* both had high expression throughout the moult cycle, with higher expression of *Shed F* in postmoult and higher *Shed C* expression in intermoult. *Shed B* has elevated expression in premoult, whilst *Shed E* is expressed in late postmoult, suggesting that *Shed* transcripts have evolved to have varied functions in different tissues and life stages in *P. ornatus*. It remains to be elucidated which gene is expressed in which type of cell in the hepatopancreas, which may assist in elucidating a specific function.

Ecdysteroidogenesis is well-characterised as it is conserved across arthropods responsible for their unique growth mechanism. However, despite the similarities of the moulting pathway among ecdysozoa, there are underlying differences between panarthropods and other ecdysozoa [58]. Four CYP450 transcripts demonstrate significantly higher expression in comparison to the rest of the *P. ornatus* repertoire. The four transcripts are all *Shed* transcripts, typically associated with the final conversion of 20HE at the target tissues; therefore, it was expected they would be expressed predominantly in the epithelial tissues. Two expression patterns were obvious within the four transcripts. *Shed C* and *Shed E* demonstrate expression primarily in the phyllosoma, with a peak between stages 11.1 and 11.2.6 for *Shed C* and a peak at gut retraction for *Shed E*. *Shed B* and *Shed F* demonstrate contrasting expression across metamorphosis, with minimal expression through the phyllosoma stages. *Shed B* expression increases at the H-phase puerulus and peaks at four days post transition into the juvenile. *Shed F* is only expressed in juvenile, significantly increasing four days post-juvenile transition. Interestingly, the hepatopancreas is not the primary source of these enzymes, as previously suggested [59]. Looking at the tissues, there is broad expression of the genes coding for enzymes across the 52 tissue libraries from juvenile and adult *P. ornatus*. The four *Shed* transcripts with elevated expression across metamorphosis also had the highest expression across the adult tissues with two expression patterns. Whilst *Shed C* shared a similar expression profile across metamorphosis with *Shed E*, across the adult tissues, *Shed C* and *Shed B* have similar expression whilst *Shed E* shares an expression profile with *Shed F*. *Shed B* and *Shed C* both have broad expression across tissues, except for muscle and hemolymph, and are the only Shed transcripts with expression in the epithelium. *Shed E* and *Shed F* have contrasting expression, primarily in the hepatopancreas. This overlap in expression between the four shed transcripts indicates that they have different functions within the adult and suggests that if we could map the individual tissues within the larval stages, it is likely that the transcripts would also display differential expression across tissues. This indicates that the functions of the *shed* gene family extend well beyond its involvement in moult. Six *Shed* orthologs were also described in *G. lateralis*, with differing expression across the hepatopancreas and other adult tissues [57]. *P. ornatus* have evolved several orthologs of *Shed* with overlapping expression and likely overlapping functions. It is important to note that aquatic invertebrates, crustaceans in particular, are known to have very little CYP450 activity in comparison to aquatic vertebrates [60]. This does not mean that they do not have important roles in crustaceans; it may mean that they require less of the active enzyme to achieve the desired outcome such as moult or nutrition metabolism. This increases the challenge to identify CYP450s of interest in crustaceans as they may have relatively little expression but still play a significant role.

Whilst 28 CYP450s were putatively annotated, 14 *P. ornatus* CYP450s remain without annotation. It is known CYP450s have a wide variety of functions in the metabolism of a variety of substrates [61]. As such, it is likely that the uncharacterised CYP450s have key roles across molecular pathways and are possibly also involved in steroidogenesis in *P. ornatus*. The 14 uncharacterised CYP450 transcripts were mapped across the available *P. ornatus* transcriptomic libraries to elucidate their potential functions. Of the 14 uncharacterised CYP450s, 12 display differential gene expression across *P. ornatus* life stages and tissues. Five of the transcripts have a close match for CYP49a1, indicating this enzyme plays a key role in *P. ornatus*. CYP49a1 is said to be an ecdysone responsive gene, with expression in the Y-organ and ovary, and to a lesser extent, the heart [62]. It has previously been suggested that its expression is likely dependent on the developmental stage in arthropods, with upregulation in early premoult animals, with expression similar to *Spook* and *neverland* in the blue crab, *Callinectes sapidus* [62]. In the adult tissues, only three of the uncharacterised CYP450s showed differential expression, all three putative CYP49a1. Two showed similar expression levels, with upregulation in the antennal gland, hepatopancreas, gills, epidermis, and fat, while the third had lower expression levels with upregulation in the neural tissues. CYP49a1 also demonstrated higher expression in the juveniles across metamorphosis. As this enzyme has differential expression across a range of stages and tissues in *P. ornatus*, it is likely to have important functions. It is part of the mitochondrial clan, along with CYP450s involved in ecdysteroidogenesis, so it is potentially implicated in steroid metabolism in *P. ornatus*, which correlates to its expression in the antennal gland and epidermis in the adults.

## 4. Materials and Methods

### 4.1. Transcriptomic Databases

There are several established transcriptomic libraries currently available for *P. ornatus*, detailing the complex life stages and tissues: an embryo development transcriptome of eleven stages during the 30-day embryo development [32]; a twelve-stage transcriptome describing the metamorphic life stages of phyllosoma, puerulus and juvenile [33]; hepatopancreas from juvenile *P. ornatus* across six stages of a complete moult cycle (Chapter 4); and multiple tissues derived from immature and adult *P. ornatus* from males and females [34]. The tissues used include eyestalk, brain, thoracic ganglia, antennal gland, testis, sperm duct, ovary, oviduct, hepatopancreas, gills, heart, stomach, intestine, muscle, epidermis, fat and hemocytes. The FASTQ files are available in the NCBI SRA database (PRJNA761502, PRJNA877712, PRJNA903480) and can also be found on CrustyBase.org [48]. All four transcriptomic libraries were screened for CYP450s.

### 4.2. Annotation of CYP450 Orthologs

A CYP450 PFAM domain search across putative proteins predicted from the *P. ornatus* metamorphosis transcriptome [33] retrieved the putative CYP450 transcripts. Following filtering and the removal of redundancies (100 percent identical transcripts at the protein coding sequence), only CYP450s encoding a minimum of 300 amino acids were kept. The amino acid sequences deduced from *P. ornatus* putative CYP450s and annotated CYP450s from arthropods were aligned via a ClustalW multiple sequence alignment using MEGA (7.0.26). A neighbour-joining tree was built with a bootstrap analysis of 1000 replicates to establish confidence in the position of the branches. Several groups of CYP450s, including the Halloween gene orthologs, were resolved through further subgroup phylogeny. The neighbour-joining tree resolved nine clusters and was used to confirm the identity of the CYP450 encoding transcripts of *P. ornatus*. The protein sequences derived from the *P. ornatus* metamorphosis transcriptome were used to identify the *P. ornatus* equivalent sequences in the other *P. ornatus* transcriptomes via a tBLASTN search.

### 4.3. Gene Expression Analysis

Digital expression patterns of the *P. ornatus* CYP450s were plotted against 11 stages of embryogenesis, 12 distinct stages across metamorphosis, hepatopancreas samples from 6 stages across a juvenile moult cycle and tissues from male and female juveniles and adults, represented as reads per kilobase per million reads (RPKM). This highlighted CYP450 transcripts identified as differentially expressed genes. An RNA-seq experiment was set up in QIAGEN CLC Genomics Workbench (11) using the CYP450 sequences to map their differential expression across 11 stages of embryogenesis (*n* = 3 per stage), 12 metamorphic stages (*n* = 3 per stage), 52 tissue libraries from juvenile and adult *P. ornatus* as well as in the hepatopancreas in 6 stages across the moult cycle of *P. ornatus* juveniles (*n* = 3 per stage).

## 5. Conclusions

CYP450s have essential roles in the functioning of crustaceans, with conserved CYP450s responsible for their growth mechanism through moulting. As CYP450s are such a diverse superfamily, they are likely to have roles in other important growth and development pathways. A total of 42 putative CYP450s were identified in *P. ornatus*, with 28 putatively annotated through phylogeny. Of the 28 annotated, 15 are involved in ecdysteroidogenesis, including the well-conserved Halloween genes. Four *Shed* transcripts demonstrated distinct differential expression that overlapped between the metamorphosis and adult life stages. This indicates that six *Shed* orthologs differ in their roles according to the life stage and tissue they are expressed in, and potentially are involved in pathways outside of their designated roles in converting ecdysone into 20HE. As CYP450s are important genes for a variety of processes associated with growth, moult and reproduction, it is likely that the remaining unannotated transcripts may have essential roles and functions for *P. ornatus*, such as involvement in the elusive “Black box” that could be resolved with further investigation.

## Figures and Tables

**Figure 1 ijms-25-01070-f001:**
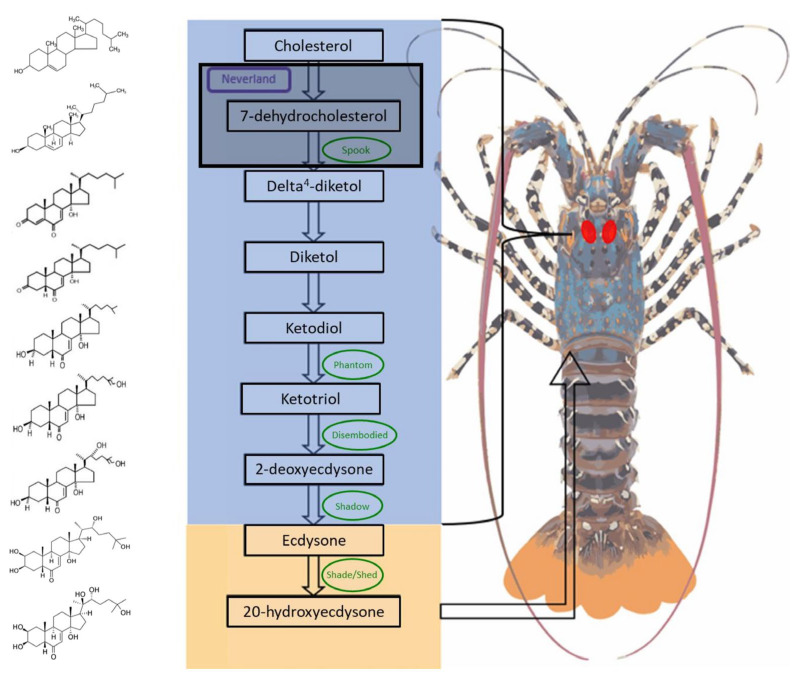
The biosynthetic pathway of the active moult hormone, 20-hydroxyecdysone, in *Panulirus ornatus*, highlighting the molecular structure changes (left) through a series of chemical reactions catalysed by Cytochrome P450s (outlined in green). The majority of the steps in this pathway (blue box) occur predominantly within the Y-organs (endocrine glands depicted by red marks) which is homologous to the prothoracic glands of insects. The initial reactions are synthesized by unknown enzymes, thus named the “black box” (outlined in black). The final conversion to the active moult hormone, 20-hydroxyecdysone (orange box), takes place in the target tissues.

**Figure 2 ijms-25-01070-f002:**
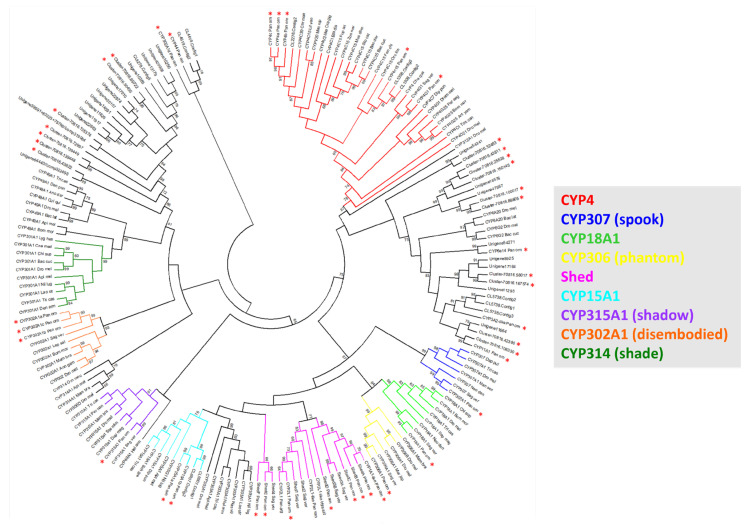
*Panulirus ornatus* Cytochrome P450 phylogeny with CYP450s from representative arthropod species, including the closely related eastern spiny lobster, *Sagmariasus verreauxi*. CYP450 protein sequences were used to form the circular cladogram showing clusters of related CYP450s across arthropods. The highlighted clades represent those containing annotated CYP450 transcripts from *P. ornatus* (marked with red asterisks).

**Figure 3 ijms-25-01070-f003:**
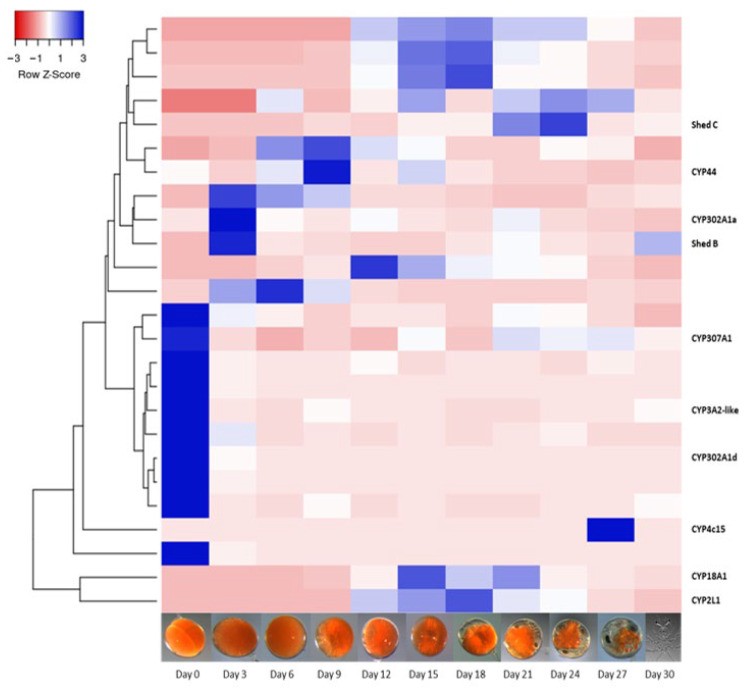
Heatmap of 25 putative Cytochrome P450s identified in embryos across 11 stages of embryonal development in *Panulirus ornatus*, created using Heatmapper [40]. Annotated CYP450s are listed by name (refer to Table 1).

**Figure 4 ijms-25-01070-f004:**
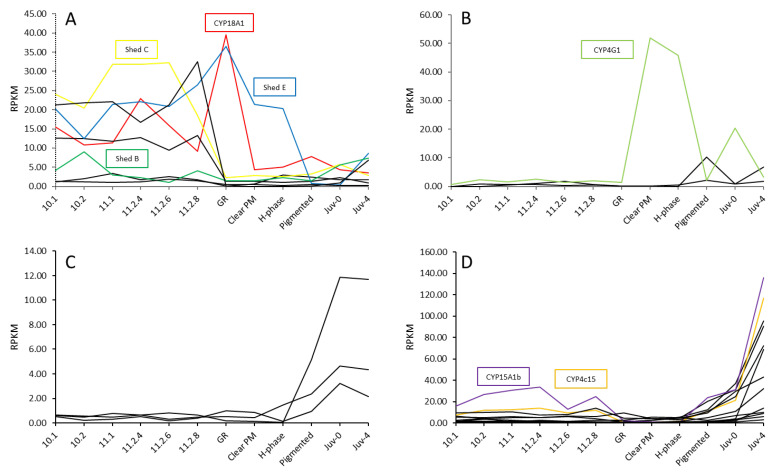
Twenty-seven Cytochrome P450s with differential expression across the 12 life stages across metamorphosis sampled from the ornate spiny lobster *Panulirus ornatus*, grouped into four expression patterns. Upregulation in phyllosoma (**A**); Upregulation in puerulus (**B**); Upregulation in Juv-0 (**C**); Upregulation in Juv-4 (**D**). Annotated CYP450s are listed by name (refer to Table 1). Twelve stages include seven phyllosoma stages; 10,1, 10.2, 11.1, 11.2–4d, 11.2–6d, 11.2–8d concluding with gut retraction (GR), three puerulus stages clear postmoult (Clear PM), H-phase, and pigmented, and two juvenile stages immediately following the moult event (Juv-0) and 4 days postmoult (Juv-4). Labels with −*n* denotes *n* days postmoult, and *n.x.* denotes stage *n*, instar *x*. Coloured lined denote annotated CYP450s, black lines represent CYP450s without annotation.

**Figure 5 ijms-25-01070-f005:**
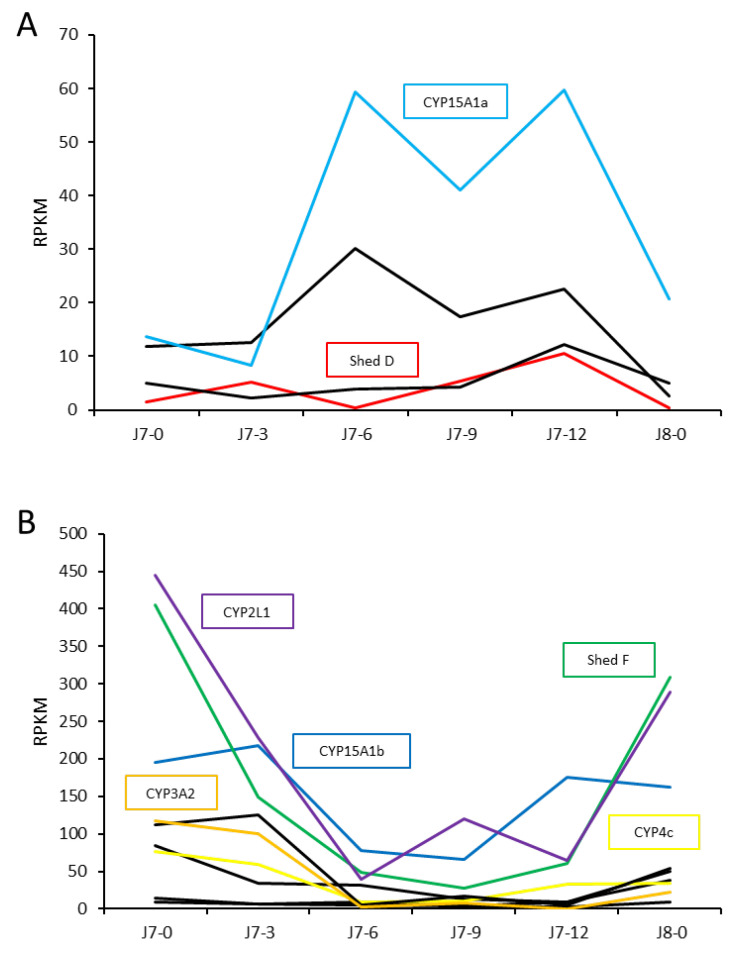
Cytochrome P450 transcript expression in the hepatopancreas from the seventh to eighth moult as a juvenile, showing transcripts with higher intermoult (**A**) and postmoult (**B**) expression. Annotated CYP450s are listed by name (refer to Table 1). J7-0: postmoult J7. J7-3: 3 days following ecdysis. J7-6: 6 days following ecdysis. J7-9: 9 days following ecdysis. J7-12: 12 days following ecdysis. J8-0: postmoult J8. Coloured lined denote annotated CYP450s, black lines represent CYP450s without annotation.

**Figure 6 ijms-25-01070-f006:**
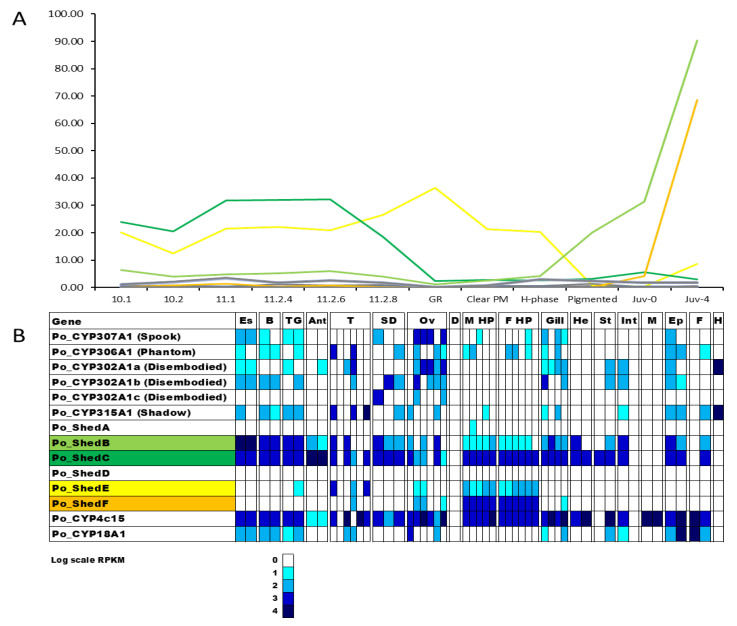
Expression of ecdysteroidogenesis-associated Cytochrome P450s across metamorphosis (**A**) and adult tissues (**B**) in *Panulirus ornatus*. Twelve stages include seven phyllosoma stages concluding with gut retraction (GR), three puerulus stages clear postmoult (Clear PM), H-phase, and pigmented, and two juvenile stages immediately following the moult event (Juv-0) and four days postmoult (Juv-4). Adult tissues are from one male (first box) and females (second box), with the exception of testes (three mature males and three immature males), sperm duct (three regions of sperm duct from one male), ovary (three mature females and three immature females), oviduct (one female), hepatopancreas (five males and six females), hemolymph (one male). Es: eyestalk, B: brain, TG: thoracic ganglia, Ant: antennal gland, T: testis, SD: sperm duct, Ov: ovary, M Hp: male hepatopancreas, F Hp: female hepatopancreas, He: heart, St: stomach, Int: intestine, M: muscle, Ep: epidermis, F: fat, H: hemolymph. Coloured lines (**A**) represent four individual CYP450s with highlighted expression adult tissues (**B**).

**Figure 7 ijms-25-01070-f007:**
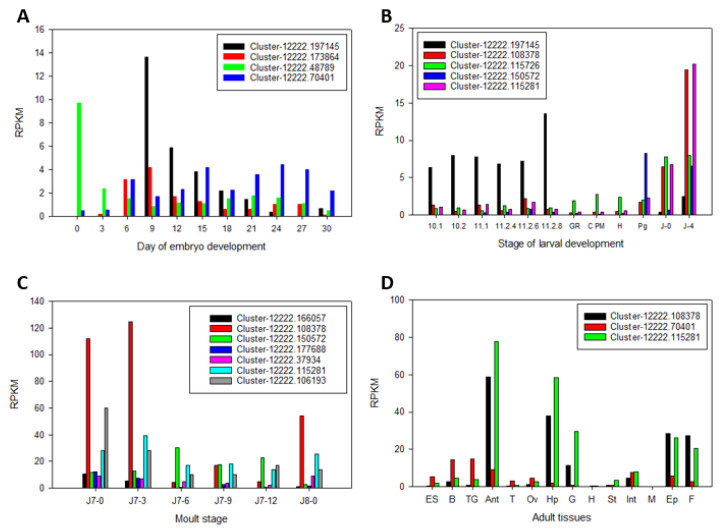
Unannotated Cytochrome P450 expression across 11 stages of embryo development (**A**), 12 stages of larval development (**B**), hepatopancreas across 6 stages of the juvenile moult cycle (**C**), and 14 tissues from male and female adult *Panulirus ornatus* (**D**). ES: eyestalk, B: brain, TG: thoracic ganglia, Ant: antennal gland, T: testis, Ov: ovary, Hp: hepatopancreas, H: heart, St: stomach, Int: intestine, M: muscle, Ep: epidermis, F: fat.

**Table 1 ijms-25-01070-t001:** Annotated Cytochrome P450s from ornate spiny lobster, *Panulirus ornatus*, listed in order of their associated pathways and clans. *Pan_orn* = *Panulirus ornatus*.

Transcripts	Annotation	Associated Pathway	Clan
Cluster-12222.190192	CYP307A1_*Pan_orn* (*Spook*)	Ecdysteroidogenesis	CYP2
Cluster-12222.47152	*Shed A*_*Pan_orn*	Ecdysteroidogenesis	Mitochondrial CYP
Cluster-12222.105348	*Shed B*_*Pan_orn*	Ecdysteroidogenesis	Mitochondrial CYP
Cluster-12222.144497	*Shed C*_*Pan_orn*	Ecdysteroidogenesis	Mitochondrial CYP
Cluster-12222.19622	*Shed D*_*Pan_orn*	Ecdysteroidogenesis	Mitochondrial CYP
Cluster-12222.37028	*Shed E*_*Pan_orn*	Ecdysteroidogenesis	Mitochondrial CYP
Cluster-12222.37028	*Shed F*_*Pan_orn*	Ecdysteroidogenesis	Mitochondrial CYP
Cluster-12222.26280	CYP315A1_*Pan_orn* (*shadow*)	Ecdysteroidogenesis	Mitochondrial CYP
Cluster-12222.74444	CYP302A1a_*Pan_orn* (*disembodied*)	Ecdysteroidogenesis	Mitochondrial CYP
Cluster-12222.32150	CYP302A1b_*Pan_orn* (*disembodied*)	Ecdysteroidogenesis	Mitochondrial CYP
Cluster-12222.175902	CYP302A1c_*Pan_orn* (*disembodied*)	Ecdysteroidogenesis	Mitochondrial CYP
Cluster-12222.184161	CYP302A1d_*Pan_orn* (*disembodied*)	Ecdysteroidogenesis	Mitochondrial CYP
Cluster-12222.93853	CYP4c15_*Pan_orn*	Ecdysteroidogenesis	CYP4
Cluster-12222.171626	CYP306A1_*Pan_orn* (*phantom*)	Ecdysteroidogenesis	CYP2
Cluster-12222.113231	CYP18A1_*Pan_orn*	Ecdysteroidogenesis	CYP2
Cluster-12222.52575	CYP15A1a_*Pan_orn*	Juvenile hormone biosynthesis [36]	CYP2
Cluster-12222.175641	CYP15A1b_*Pan_orn*	Juvenile hormone biosynthesis	CYP2
Cluster-12222.93855	CYP4G1_*Pan_orn*	Lipid metabolism	CYP4
Cluster-12222.55035	CYP4a_*Pan_orn*	Omega hydroxylation [37]	CYP4
Cluster-12222.52849	CYP4b_*Pan_orn*	Omega hydroxylation	CYP4
Cluster-12222.55185	CYP4c_*Pan_orn*	Omega hydroxylation	CYP4
Cluster-12222.102364	CYP2L1-like_*Pan_orn*	Estrogen synthesis	CYP2
Cluster-12222.82549	CYP3A2-like_*Pan_orn*	Oxidative metabolism [38]	CYP3
Cluster-12222.159369	CYP6a14_*Pan_orn*	Hormone biosynthesis	CYP3
Cluster-12222.134345	CYP44_*Pan_orn*	Oxidative metabolism	Mitochondrial CYP
Cluster-12222.139601	CYP1A1-like_*Pan_orn*	Xenobiotic metabolism	CYP2
Cluster-12222.56205	CYP11A1a_*Pan_orn*	Steroidogenesis [39]	Mitochondrial CYP
Cluster-12222.166057	CYP11A1b_*Pan_orn*	Steroidogenesis	Mitochondrial CYP

**Table 2 ijms-25-01070-t002:** Unannotated Cytochrome P450s with differential expression identified across the available transcriptomic databases for *Panulirus ornatus*: 11 stages across embryo development, 12 stages across larval development, the hepatopancreas across 6 stages of the moult cycle in juveniles, and multiple tissues from male and female adults. Hp: hepatopancreas. Blue shading indicates differential expression of the unannotated CYP450.

	Embryo	Larval	Juvenile Hp	Adult
Cluster-12222.166057				
Cluster-12222.108378				
Cluster-12222.150572				
Cluster-12222.177688				
Cluster-12222.37934				
Cluster-12222.115281				
Cluster-12222.106193				
Cluster-12222.70401				
Cluster-12222.197145				
Cluster-12222.173864				
Cluster-12222.48789				
Cluster-12222.115726				

## Data Availability

The data presented in this study are openly available in the NCBI SRA database (PRJNA761502, PRJNA877712, PRJNA903480) and can also be found on CrustyBase.org, accessed on 9 January 2024.

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
