# Peer review of "Ontogeny of the Cytochrome P450 Superfamily in the Ornate Spiny Lobster (*Panulirus ornatus*)"

_ijms, 2024, doi:10.3390/ijms25021070_

Round 1
Reviewer 1 Report
Comments and Suggestions for Authors
In this manuscript, Lewis et al. have identified the transcripts putatively encoding for CYP450s and analyzed the expression patterns of these transcripts across growth and development stages of P. ornatus. This study provides insights into further functionally characterization of CYP450s in this species. The manuscript is well-written and follows a clear flow of thoughts. The figures are informative and presented clearly. I recommend publication of the present manuscript and I have a few suggestions for the authors on how to further strengthen it.
1. The authors determined the expression patterns of these putative genes from a transcriptome data of P. ornatus. Are these transcriptome data published or available in NCBI? The authors did not perform any experiments to confirm the RNA-seq results. I would suggest the authors choose some target genes and verify the RNA-seq results using RT-PCR.
2. In Figure 4 and Figure 5, what are the meanings of the black line?
Author Response
We thank the reviewer for their efforts.
Our response to the feedback is below.
- The transcriptome data is published and available on NCBI. The three separate publications (where the embryos, larvae+juveniles and adult tissues were sequenced) were referenced in the text and the accession numbers for the databases are available in these three publications. The data can also be accessed through CrustyBase.org - a dedicated server for crustacean transcriptomic gene expression analysis. We no longer have access to the RNA samples to run qPCR or RT-PCR validations. The expression patterns of key ecdysone synthesis genes is consistent with literature from other species, strengthening the validity of our RNA-Seq data. Moreover, there are at least 20 publications from our team, where we performed qPCR analysis to validate the data from the transcriptomes explored in this research.
2. In Figure 4 and Figure 5, what are the meanings of the black line?
Black lines represent un-annotated CYP450-encoding transcripts.
Reviewer 2 Report
Comments and Suggestions for Authors
Dear authors, here is my review of the manuscript entitled "Ontogeny of the cytochrome P450 superfamily in the ornate spiny lobster (Panulirus ornatus)".
After carefully reading the manuscript, I can assess I did not identify any flaws.
In my opinion, the manuscript is suitable for publication as it is.
Best regards.

Author Response
We thank the reviewer for his efforts.